# Specifying Spatial Dependence for Teak Stands Specific to Solomon Island-Derived Clones in Tawau, Sabah, Malaysia: A Preliminary Study

**Johannah Jamalul Kiram** [1,2], **Rossita Mohamad Yunus** [2,*], **Yani Japarudin** [3] **and Mahadir Lapammu** [3]

1    Preparatory Centre for Science and Technology, Universiti Malaysia Sabah, Jalan UMS, Kota Kinabalu 88400, Malaysia; johannah612@ums.edu.my
2    Institute of Mathematical Sciences, Faculty of Science, Universiti Malaya, Kuala Lumpur 50603, Malaysia
3    Sabah Softwood Berhad, Tawau 91000, Malaysia; yanijaparudin@yahoo.com (Y.J.); mahadir_ssb@yahoo.com (M.L.)
*    Correspondence: rossita@um.edu.my

**Abstract:** The magnitude of spatial dependence on teak tree growth was examined based on a teak plantation managed by the research and development team at Sabah Softwood Berhad, Brumas camp, Tawau, Sabah, Malaysia. A sample of 432 and 445 georeferenced individual tree points specific to Solomon Island-derived clones that were 6 and 7 years old, respectively, were analyzed, as previous findings showed that this was the genotype that thrived the most. This study aims to show that spatial dependence exists in the 6- and 7-year-old teak tree blocks of the plantation and that there are changes in the magnitude of spatial dependence when it is analyzed as a continuous plot. Moran's I values and Moran scatterplots as well as semivariograms and thematic maps were used to satisfy the hypothesis regarding the relationship between spatial dependence and the growth of the physical parameters: the diameter at breast height (DBH), height, and the volume of the teak tree. The Moran's I values that were calculated rejected the null hypothesis, suggesting the existence of strong spatial dependence for all of the physical parameters and for both the 6- and 7-year-old samples. The semivariograms were plotted and showed an increasing trend as the lag distance between trees increased and showed changes as the trees aged. These findings prove significant spatial dependence in the growth of the physical parameters of teak trees. Hence, growth model methodologies based on spatial distribution must be developed to further understand the spatial distribution of teak tree plantations.

**Keywords:** spatial dependence; Moran's I; semivariograms; teak; Solomon clone

## 1. Introduction

The growth and yield of a tree are subject to the tree's surroundings. This includes how trees are planted, their topological conditions, the chemical attributes of the soil, and the climate. Planting the same seed in two different spots, let alone in different countries on different continents, will yield different outcomes. Thus, there is a need to analyze spatial dependence regardless of what precedes growth. Spatial dependence refers to the degree of the spatial autocorrelation between the independently measured values observed in geographical space [1]. Among the obvious measurements that can be statistically made is the management of the plantation site, a variable that contributes to whether there are negative dependencies or positive dependencies, where pairs of locations that are a certain distance apart that have random variables with similar values are positive dependencies and negative dependencies otherwise [2,3]. Thus, analyzing the spatial dependence is crucial as a rudimentary step to identify variables that affect a tree's growth directly.

From a methodological viewpoint, be it in a theoretical or policy framework, spatial dependence is important when it is part of a model [4]. The main reason why a tree's growth is affected is because of the competition surrounding it. The more trees being

planted in a particular site, the more the competition intensifies, generating negative spatial dependence between interacting individuals [5]. However, studies have further observed and proven that this period of intense competition will pass and natural mortality or artificial thinning will reduce the competitive effect, hence making the attributes of an individual tree return to being an expression of micro-site influences [5]. For these reasons, a matured tree is often observed to have positive spatial dependence. However, there should be a scientific way to maximize yield in spite of positive spatial dependence before a tree reaches maturity. Analyzing spatial dependence helps us understand and appreciate the spatial distribution of the area of study, allowing us to develop an accurate model to estimate future outcomes and execute kriging.

This study examines Tectona grandis (Teak), particularly Solomon Island-derived clones grown in Tawau, Sabah. The teak tree itself has quite a reputation in the literature, as all aspects of the teak tree have been studied in situ and in vitro, in natural stands and on plantations, and all around the world, in countries such as India, Nepal, Thailand, and Brazil [6–10]. However, despite the wide range of studies, the field of spatial statistics is limited [9]. Thus, not many works of literature investigate the possibilities of whether there is any spatial dependence determining teak tree growth and, more specifically, the growth of teak trees planted in Sabah. Consequently, there is a significant need to conduct research with the primary aim of identifying the effects of spatial dependence on teak tree growth.

Focusing on studies about the teak tree itself, there is research in the literature that involves spatial modeling; however, these studies model the chemical attributes of the soil in *Tectona grandis* stands in Brasnote, Mato Grosso state, Brazil [11]. Another study models the crown ratio for the Tectona grandis, which is the ratio of the crown length to the tree height. However, the study does not consider any other spatial attributes [12]. Volume models for *Tectona grandis* can be found in a number of studies; the ones mentioned in this study were mainly conducted in Tanzania, Nigeria, and Brazil [13–16]. A study on the nutritional diagnosis of teak leaves in the eastern Amazon [17] had a similar flow as this current study, as it analyzed the spatial dependence between the nutrition in teak tree leaves and spatial data to understand and elaborate upon the variability in the map where these teak trees are being planted. However, this literature reported how growing rates, management regimens, and production objectives are specific to each country or world region. Volume equations should not be generalized assuming that trees have similar bole shapes, as quoted by Tewari and Singh [6] in their study presenting teak tree growth in India. Furthermore, this limited number of studies about teak did not include the effects of spatial or temporal data.

Popular timber supplies come from teak trees, and these materials contribute to Sabah's economy, as there is a constant demand for these resources all around the world to produce quality furniture and for construction projects. While the need for timber continues to rise, it often coincides with the subject of ecological forestry, as it markedly contributes to deforestation and reforestation. Adams [18] reported a huge decline in the world's forest cover and noted that if we combined the measurements of deforestation happening throughout the world, then the rate of forest cover decrease can reach up to 58 thousand square miles a minute. This is equivalent to at least 48 football fields. Not only does deforestation cause climate change, it also causes other environmental concerns such as water cycle changes, losses in biodiversity, and soil erosion, which are some of the few changes mentioned in the study by Bennett [19]. Thus, the need for prompt reforestation that does not disrupt the supply of priced timber is crucial to meet the requirements of constant development. This stresses the importance of improving existing statistical findings about teaks in Sabah and to aid and inspire teak plantations around the world. As previously discussed, ecologists are facing challenges in understanding and predicting the dynamics and productivity of the world's tropical rain forests [20].

The teak plantation in Sabah was planted based on research instigated by Innoprise Corporation Sdn Bhd when considering whether to invest in the mass cloning of teak trees back in 1994, and this research was conducted within the framework of a collaboration

between the CIRAD Forestry Department and ICSB [21]. ICSB then furthered their research and investment by planting a monoclonal block in 1997 on a 1.2 ha plot using microcuttings in the area [22]. The success of these monoclonal blocks inspired another two provenance cum progeny trials, which were set up in 1997 according to partially balanced incomplete block designs in two different locations in Sabah. Goh and Monteuuis [22] listed the detailed origins of the diverse collection of teak trees. There have been reports published regarding its field performance [20,21]; however, these reports did not consider its spatial variability and spatial dependence on tree parameters such as height, diameter at breast height (DBH), and volume. The bole volume formula derived from previous literature [23–25] with similar objectives can be seen in Equation (1).

$$V = \frac{1}{10}\left[\left[1.3\pi\left(\frac{D}{2}\right)^2\right] + \left[\pi\left(\frac{D}{2}\right)^2\left(\frac{H-1.3}{3}\right)\right]\right], \tag{1}$$

where *V* is the volume given in *D*, which is the DBH (1.3 m above ground), and *H* is the height of the tree.

The collaboration then extended their research, and a number of teak plantations were planted in multiple districts in Sabah, some of which were solely for research purposes, while some were for yielding teak. The data in this study were obtained from the teak plantations that are carefully managed by a research team at Sabah Softwood Berhad at the Brumas camp in Tawau district, Sabah, East Malaysia. Previous attempts to study these data did not take into account the spatial data of the plantation [21–25] and ergo the need to apply geostatistical techniques to help improve the findings and manage the research. As stated in previous studies, spatially continuous data are important in all ecosystems, including in decision making for forests [26] for autocorrelation in particular, which provides a means of characterizing spatial structure, yet different kinds of structure are revealed in different spaces [27].

The objectives of this study were to determine the spatial dependencies of the physical parameters of *Tectona grandis* that are specific to the Solomon Island-derived clone and to determine the changes in the magnitude of those spatial dependencies as the teak tree stands age. These physical parameters include tree diameter at breast height (DBH), height, and volume. We begin by identifying the relationship between the physical parameters and the spatial information of the tree. The spatial dependence test was conducted using Moran's I, and semivariograms were computed to examine the relationship between the sample tree stands. Graphical analysis using thematic maps was used to make inferential observations about the distribution of the plantation on the block that was sampled. While this study focuses on micro-sites, it helps to estimate and understand how spatial dependence will affect bigger plantations in the future.

## 2. Materials and Methods

### 2.1. Data

The seminal teak stands in this study were planted in the year 2002 and are managed by the research and development team of Sabah Softwood Berhad, Brumas camp, Tawau, Sabah, Malaysia. The camp has a number of blocks dedicated to teak plantation research that are located within the camp area. This study focuses on one block, located at the coordinates 4°37′23.85″ N and 117°47′05.12″ E. There are 12 years of data available for this plot. This study uses data from the plot's 6th and 7th years, at which point the plot covered 5.67 hectares. It was initially planned for multiple types of research, including an in vitro study on teak cloning and mass propagation using cuttings [6,19,20]. The topology map of the current site is shown below in Figure 1. The image was provided by the Sabah Softwood Berhad research team.

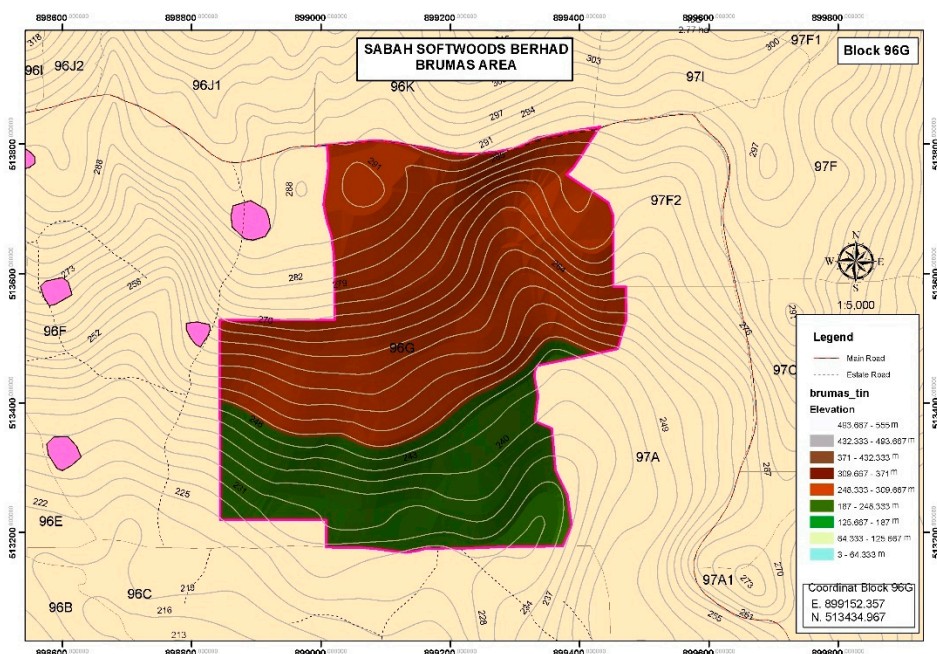

**Figure 1.** The topologic map of block 96G at Brumas Camp, Tawau, Sabah.

There are 15 different teak genotypes that are produced by micro-propagation planting [23], and seven of them are Solomon Island-derived clones. These Solomon clones seemed to thrive better than the other clones and were selected for further observations in this study.

A randomized complete block design with four contiguous replications was used. The plots comprised two rows each of 30 plants belonging to 15 different genotypes. The plots were $4 \times 4$ m$^2$ and had 625 stems per hectare, resulting in over 4000 trees. Only the 11th to 20th plants of each row were assessed, corresponding to a total of 80 plants per clone. The current block 96G is on sloped land at an altitude ranging from 180 to 370 m above sea level. The region's climate has been classified as tropical rainforest (Köppen), while its soil is classified as Tg Lipat soil with a low content of nitrogen, potassium, and magnesium.

Sample data were collected for 1200 trees, and the team recorded their heights and diameters at breast height (DBH), and their volumes were calculated using Equation (1). All of the data were collected from one to four replicates across all fifteen treatments that were administered. However, due to the specific sample requirements of this study, only Solomon Island-derived clone data are analyzed. As of the 6th and 7th year, only 801 out of the 1200 trees were still standing, not including the outliers experiencing undergrowth. Samples were collected from 451 trees that were 6 years old and 451 trees that were 7 years old. All of the statistical analyses in this study were executed using R Studio 4.0.5.

Out of these samples, 432 georeferenced individual tree points were obtained for the 6th year plot, and 445 georeferenced individual tree points were obtained for the 7th year plot. The georeferencing points were accurate to up to six decimal points. Further geostatistical analysis was carried out using ArcGIS 10.8.1 and R Studio 4.0.5.

### 2.2. Analysis Structure

The flow of the analysis must be clearly defined to ensure the smooth structure of the model and the accuracy of the analysis. All the assumptions must be considered before proceeding on to the next step. Hence, a defined algorithm must be followed to ensure optimum findings. Figure 2 shows a brief flow chart that defines our steps before we achieved the ultimate goal of this research, which was to formulate a spatial model that is able to provide accurate predictions and further understanding of teak management. Our model was inspired by Olea [28], who proposed a six-step practical approach to semivariogram modeling in geostatistics, but we altered that approach to follow the design of our experiment. We

consider this particular study a step towards identifying spatial dependence. Following this current study, we will model the spatial dependence and further estimate the possible yield to the highest degree of accuracy possible in future studies.

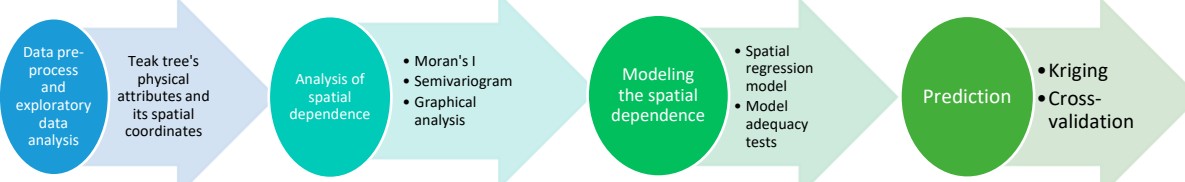

**Figure 2.** Flow chart to approach spatial modeling in managing *tectona grandis*.

The aim of this study was to focus on the analysis of spatial dependence to ensure that the data that are collected are reasonable. Studies [5,29,30] have emphasized the importance of analyzing the spatial autocorrelation. Reed and Bukhart [30] asserted that successful autocorrelation findings will lead to a more thorough understanding of the response of the stands to changes in their surroundings, such as competition in response to recorded thinning. This deep understanding of spatial dependence will aid in the development of more successful management options.

To examine the first hypothesis, there is spatial dependence in the block of the teak plantation containing 6- and 7-year-old trees. Moran's I was used on the teak tree plots for each year and was applied to every physical parameter. Since the stands are young, the results are expected to show small changes. As this study continues and as more data are collected, bigger differences are expected in future studies. The second hypothesis is that there are changes in spatial dependence when it is analyzed as a continuous plot. In this study, we observe two consecutive years of data to examine the second hypothesis and to determine whether the change shows negative or positive spatial dependence for every physical parameter. Semivariograms will be administered, as they consider the distances between individual tree stands instead of just the weight matrices, which is different from the method used in the method of the first hypothesis. Graphical analysis is performed to further solidify the numerical outcomes that are obtained.

*2.3. Defining Spatial Dependence Methods*

2.3.1. Moran's I

Moran's I is the traditional method used to calculate spatial dependence [2,24,25]. The formulation for Moran's I is

$$I = \frac{n \times \sum_{i=1}^{n} \sum_{j=1}^{k} w_{ij} (Z_i - \overline{Z})(Z_j - \overline{Z})}{\sum_{i=1}^{n} \sum_{j=1}^{k} w_{ij} \sum_{i=1}^{n} (Z_i - \overline{Z})^2} \tag{2}$$

where $n$ is the total number of samples, which in this study would be the number of sampled teak trees; $k$ is the number of neighbors; $w_{ij}$ is the weight of the interaction between trees $i$ and $j$ (and is equal to 1 for all interactions); $Z_i$ and $Z_j$ are data values for trees $i$ and $j$, respectively; and $\overline{Z}$ is the mean data value. There are no pre-defined statistical methods to determine the most favorable *k value*. However, in this study, we defined *k* using Delaunay triangulation [5]. Delaunay triangulation is a technique that is used to create a mesh of contiguous, nonoverlapping triangles to help identify near neighbors when spatial arrays are irregular [31]. To execute Moran's I, we begin by constructing a spatial weight matrix $w_{ij}$. It generalizes spatial dependence across the trees in a plot.

To further validate our methods, we proceeded by plotting Moran scatterplots [4]. Moran scatterplots provide a visual representation of the spatial associations in the neighborhood around each observation, which in this case would be each sampled tree. Thus, it provides a more detailed reasoning for spatial dependency inferencing. Moran scatterplots

have a *y*-axis that explains the spatial lag variable compared to the original variable on the *x*-axis. The lower left quarter and the upper right quarter of the plot indicate positive spatial dependencies, and the fitted line is the mean. These quarters are usually referred to as the high–high and the low–low of the spatial autocorrelation [32]. Since every individual tree point has an irregular structure, it considers the spatial lag operator, which consists of the weighted average of the values at neighboring locations [4]. The spatial lag is obtained as the product of a spatial weight matrix $w_{ij}$.

### 2.3.2. Semivariogram

Semivariograms are statistics that assess the average decrease in similarity between two random variables as the distance between the variables increases, and they have some applications in exploratory data analysis [33]. They quantify assumptions that samples in a similar location tend to behave in a similar way compared to samples that are farther away. They assess the average decrease in similarity between two random variables $Z(x_i)$ and $Z(x_i + h)$; as the distance between them increases, further applications in exploratory data analysis are possible [34]. In this context, $x_i$ and $x_i + h$ are the spatial positions separated by a vector, *h*. Semivariograms measure the spatial dependence between two observations as a function of distance. If the distance between two points *h* should not be equal to 0, then the semivariogram function will vary from more than zero until the highest *h* values, where the points will be the farthest away from each other. Since the aim of this research is to be able to make high-accuracy estimations, i.e., kriging, semivariograms are required [33]. With that being said, the nugget effect is an essential part of the model.

The nugget effect in the model is a constant value for all of the distances between two points that are greater than zero and can be estimated by extrapolating the variogram to an intercept on the variogram axis [35]. The semivariogram in Equation (3) was considered for the geographical positions of the plots and includes the lag distance and the numerical differences of each variable on the grid. This will help to interpret the spatial continuity of the data. To conduct a semivariogram, the data must be approximately normal. In the equation below, $y(h)$ is the semivariance of the $Z(x_i)$ variable, $N(h)$ is the number of pairs of plots for each lag, and *h* is the distance [33].

$$y(h) = \frac{1}{2N(h)} \sum_{i=1}^{N(h)} [Z(x_i) - Z(x_i + h)]^2 \tag{3}$$

### 2.3.3. Thematic Map

The observations based on thematic maps plotted to show the distribution of the physical parameters of the teak tree stands that were sampled. The plot was created using R Studio 4.0.5. Each physical parameter—diameter at breast height (DBH), height, and volume—were plotted for observation according to the spatial distribution based on the heat represented by the different colors. The value of the physical parameters increases in tandem with the increase in the heat represented by the color, which shifts from yellow to red. Patterns should be spotted if there is any spatial dependence. Otherwise, the plot should be random without any obvious patterns if the growth of the trees is independent of the spatial distribution.

## 3. Results

### 3.1. Moran's I Analysis

Table 1 provides the computed Moran's I values. For all of the physical parameters and both years, the Moran's I values showed a *p*-value less than 0.001. Hence, the null hypothesis can be rejected, suggesting that there is a cluster or dependence among these parameters. A Monte Carlo simulation of Moran's I with 999 simulations also suggests that there is a less than 1% chance of being wrong in rejecting the null hypothesis. The Moran's I values were not only computed to decide whether to reject or accept the null hypothesis; if the values exhibit a positive or negative spatial dependence or are equal to



zero, then there is no autocorrelation. However, if the values are −1 or 1, there is perfect clustering/dependence. However, the calculated Moran's I values in Table 1 suggest that the height of teak trees is mostly affected by the spatial attribute. This could be due to competition and the fact that it was planted on sloped land.

**Table 1.** Moran's I values and the *p*-values and Monte Carlo simulation of the Moran's I values for the physical parameters of the 6- and 7-year-old trees.

| | Moran's I | 6th Year *p*-Value | Monte Carlo | Moran's I | 7th Year *p*-Value | Monte Carlo |
|---|---|---|---|---|---|---|
| DBH | 0.2568735864 | <0.0001 | 0.001 | 0.2321702692 | <0.0001 | 0.001 |
| Height | 0.5525857451 | <0.0001 | 0.001 | 0.6033771319 | <0.0001 | 0.001 |
| Volume | 0.3853519365 | <0.0001 | 0.001 | 0.3452859439 | <0.0001 | 0.001 |

The Moran scatterplots are shown in Figures 3 and 4. These scatterplots show the relationship between the spatial lag and the teak tree's physical parameters—DBH, height, and volume—for both the 6- and 7-year-old stands. When the plots increase in line with the fitted regression line, a positive association can be observed. This can be seen in all six plots. The plot for the heights and volumes for both the 6- and 7-year-old trees shows an obvious strong relationship, as most of the plot falls near the fitted line, suggesting that there is strong spatial dependence.

However, the relationship between DBH and the spatial lag counterpart for the 6th and 7th year plots, despite suggesting a positive association, has a number of samples that fall far from the fitted line. Moreover, the shape of these two DBH plots is almost random and is roundish instead of conic compared to the plots of height and volume for both years, though from a macro point of view, the plots suggest a positive linear relationship, especially as seen in the 6th year plot for the DBH. The plot also suggests that there may be outliers. However, these outliers are not to be seen as outliers in a traditional statistical sense but as outliers relative to their spatial neighbors [36]. However, with the calculated Moran's I values in Table 1 and despite the positive linear association and the shape of the plot, we should be able to conclude that both the 6th and 7th year scatterplots for the DBH suggest that there is a relationship between the spatial lag and the DBH, hence indicating strong spatial dependence in each of the 6th year and 7th year blocks of the teak tree plantation.

### 3.2. Semivariogram Analysis for Spatial Dependence

Figures 5 and 6 are the semivariogram plots for the physical parameters of the samples taken from the 6- and 7-year-old teak trees. The *x*-axis represents the distance between the samples. The *y*-axis is the calculated semivariance, which is the measure of spatial dependency between two observations. The fitted line is the theoretical semivariogram according to spherical distribution and is used to help approximate the sill, range, and nugget of the plots. All three physical parameters show a similar increasing trend, which is caused by the nugget variance. The nugget variance is the difference between the semivariance estimates at the smallest distance class between two plots and zero. It reflects the magnitude of the random spatial variation in every attribute that is being studied in this research [5]. The trend is similar, however, as an increased difference in the values of the *y*-axis can be compared between these two years.

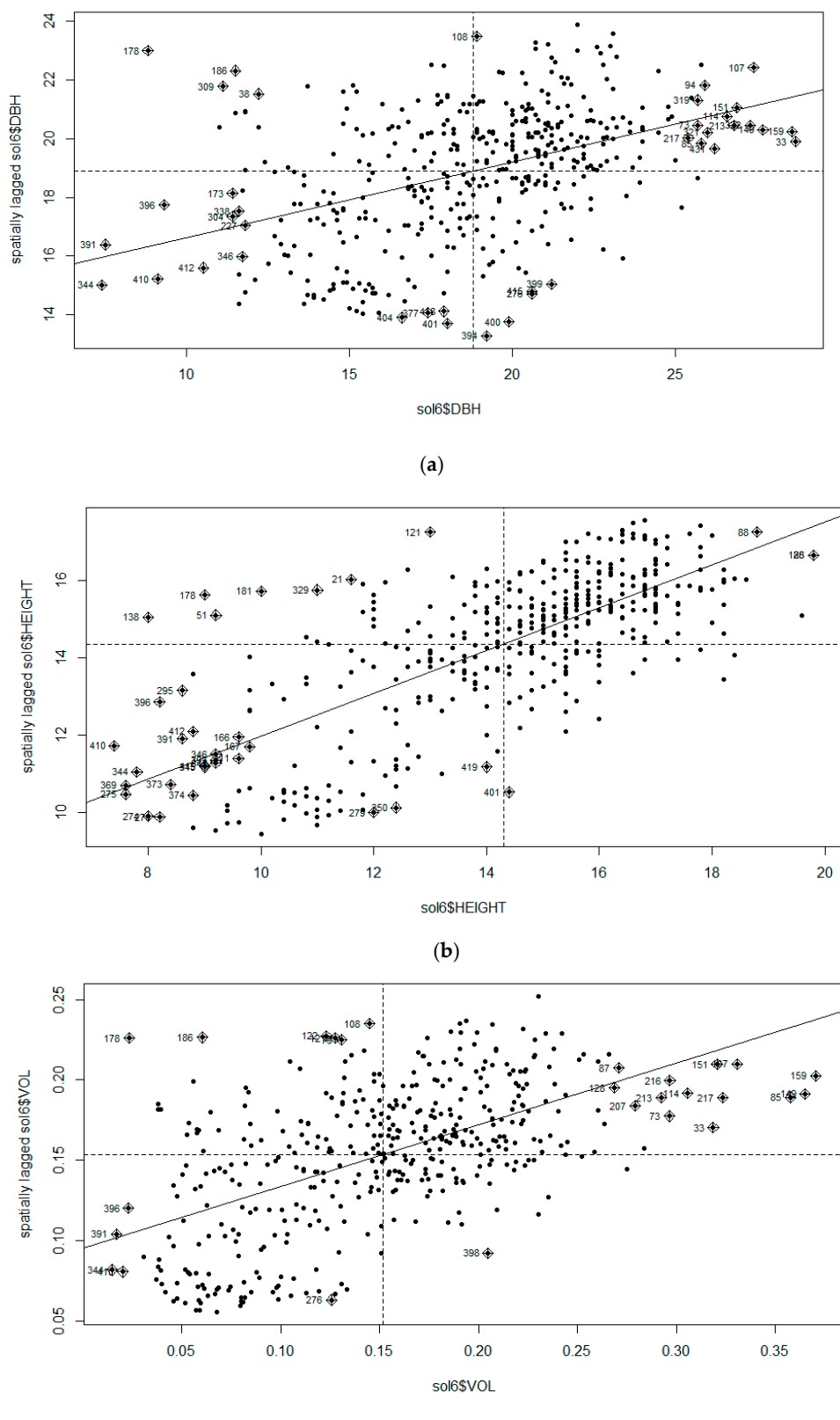

**Figure 3.** Moran scatterplot of (**a**) DBH, (**b**) height, and (**c**) volume for the 6-year-old trees.

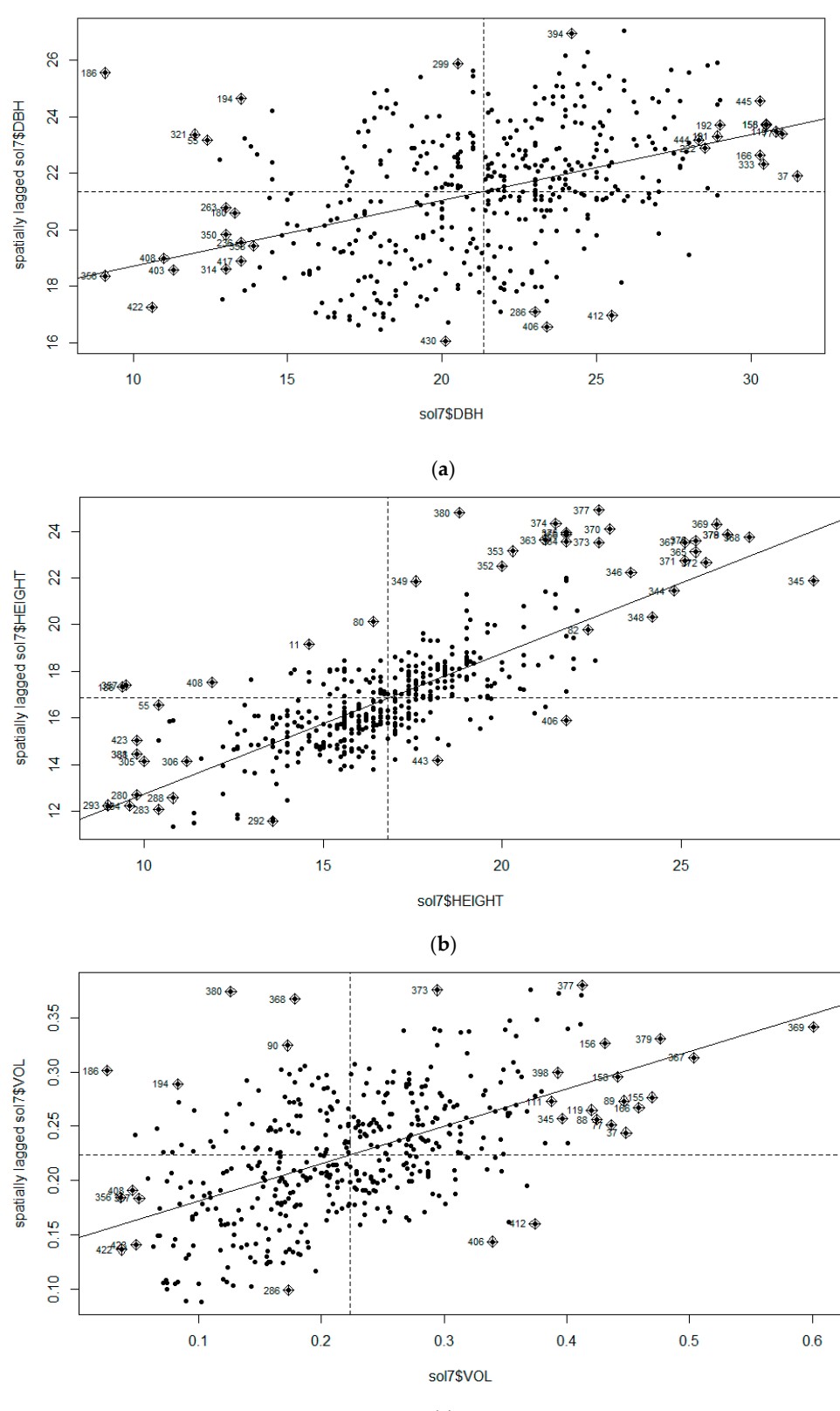

**Figure 4.** Moran scatterplots of (**a**) DBH, (**b**) height, and (**c**) volume of the 7-year-old trees.

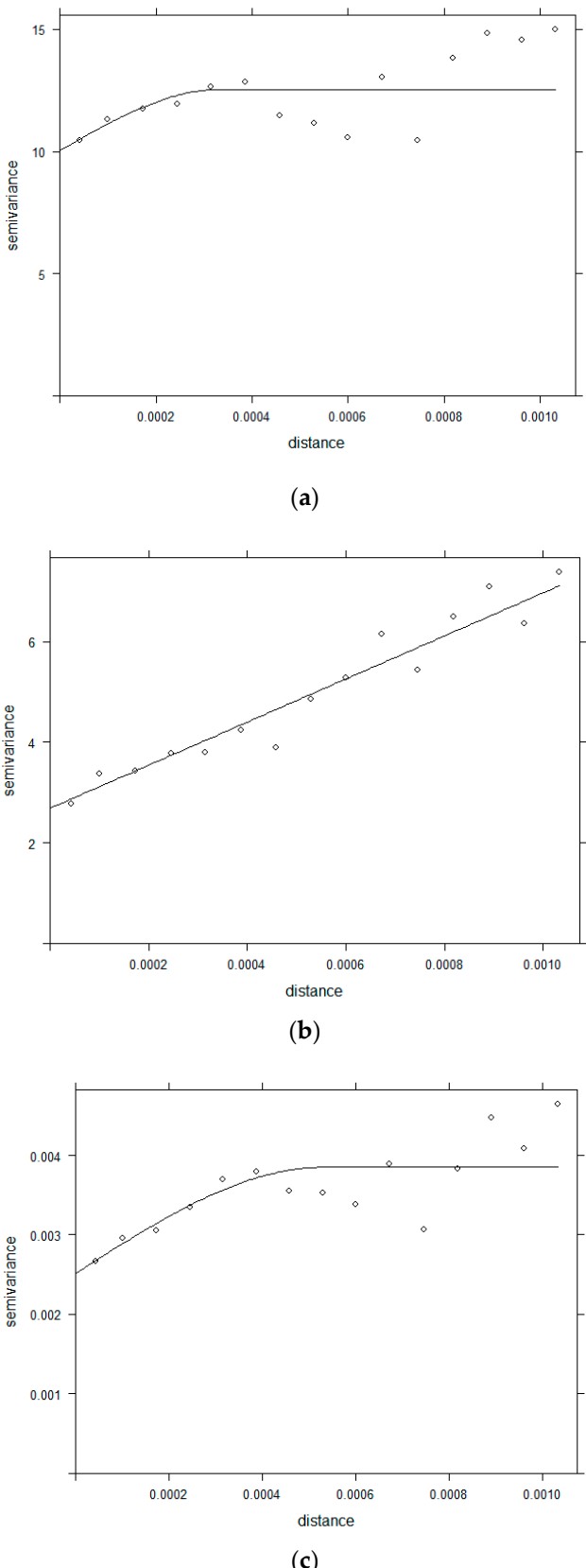

(**a**)

(**b**)

(**c**)

**Figure 5.** Semivariogram plots of (**a**) DBH, (**b**) height, and (**c**) volume for the 6-year-old trees.

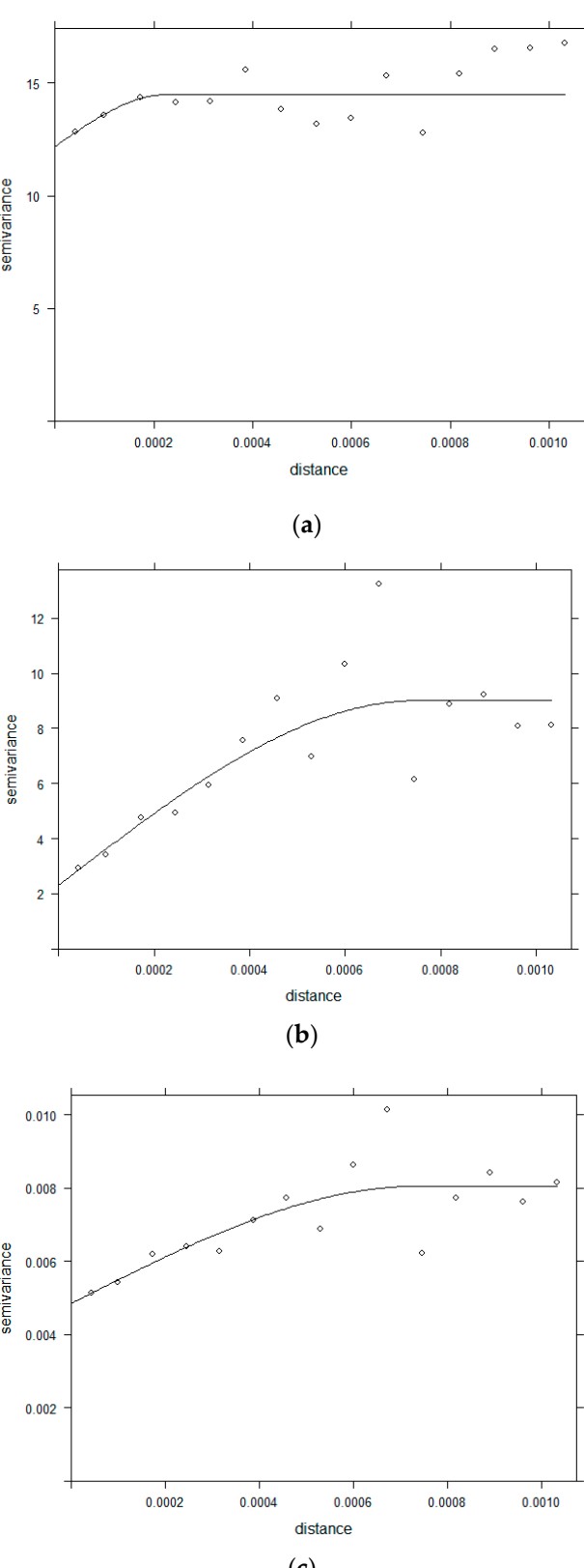

**Figure 6.** Semivariogram plots of (**a**) DBH, (**b**) height, and (**c**) volume for 7-year-old trees.

The distance unit used in these graphs is kilometers. Focusing on the *y*-axis, there is a steady increase when comparing the 6th year to the 7th year graphs. This increase is caused by increased nugget variance. The nugget reflects the magnitude of the random spatial variation,

particularly for the DBH, height, and volume of the teak tree. Here, then nugget suggests that there is an increase in the magnitude of the spatial dependency as the tree ages and grows, and the presence of spatial dependency is up to approximately 4 m, at which a sill is reached for the DBH, and at 6 m for height and 5 m for volume. However, the semivariance between the two years showed increasing values on the *y*-axis, from about 15 cm for the DBH of the 6th year to above 15 cm for the DBH of the 7th year. For the height, there was an increase from around 8 m for the 6th year to above 12 m for the 7th year. Finally, for volume, there was an increase from about 0.006 m$^3$ for the 6th year to above 0.010 m$^3$ for the 7th year. This indicates that there are changes in the spatial dependence when it is analyzed as a continuous plot.

### 3.3. The Graphical Analysis of Thematic Maps

To further solidify our hypothesis that spatial dependence exists and affects teak tree growth, we plotted thematic maps of the individual teak tree stands that were sampled based on their physical parameters. Figure 7 shows a thematic map of the parameters height, DBH, and volume for the 6-year-old individual teak tree sample stands. Similar growth can be seen towards the south-west of these three maps, where warmer colors are prevalent for all three parameters. The north side of the map shows a warmer value, too. However, the blocks in the central-east portion seem cooler than the other areas. The west side of the map shows warmer colors, indicating more fertile growth.

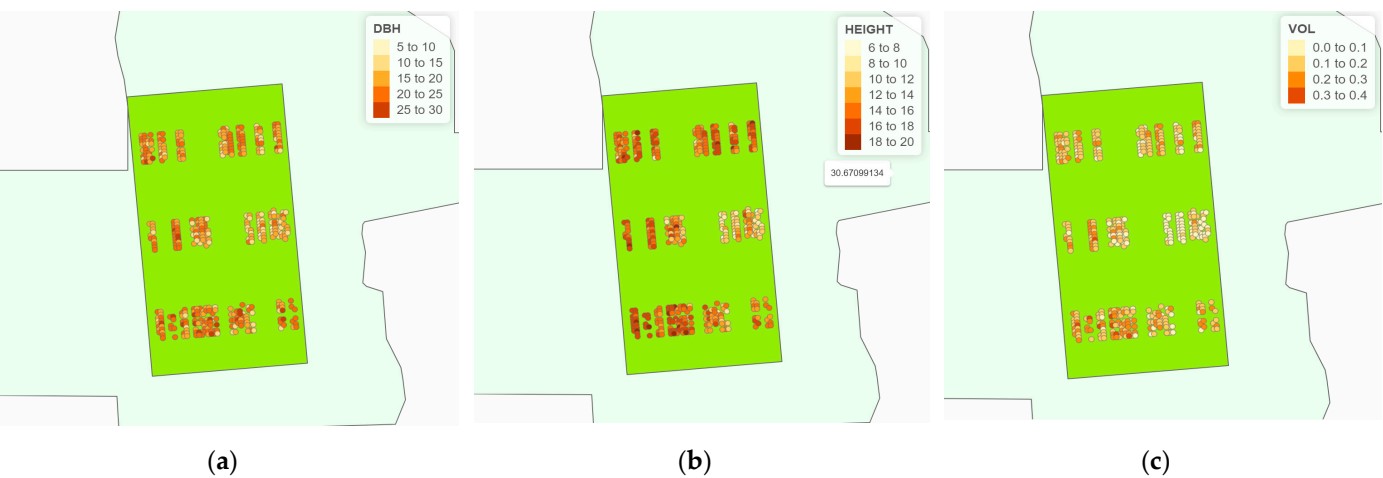

(**a**)　　　　　　　　　　　　　(**b**)　　　　　　　　　　　　　(**c**)

**Figure 7.** Sample stand spread of (**a**) DBH, (**b**) height, and (**c**) volume in the teak plots of 6-year-old Solomon Island-derived clones in block 96G at Brumas Camp, Tawau, Sabah.

A similar pattern can be observed in the 7-year-old individual teak tree sample plots in Figure 8, where warm colors are observed in the south-west and north portions, and cooler colors are observed in the central-east section of the maps. The similar patterns, disregarding the different years, suggest a spatial dependence across blocks. The growth difference was apparent in the values representing the warmer colors, but the pattern showing the density of the growth throughout the blocks is obvious in its color distribution, in which there is an autocorrelation between spatial dependency and the tree's parameter growth.

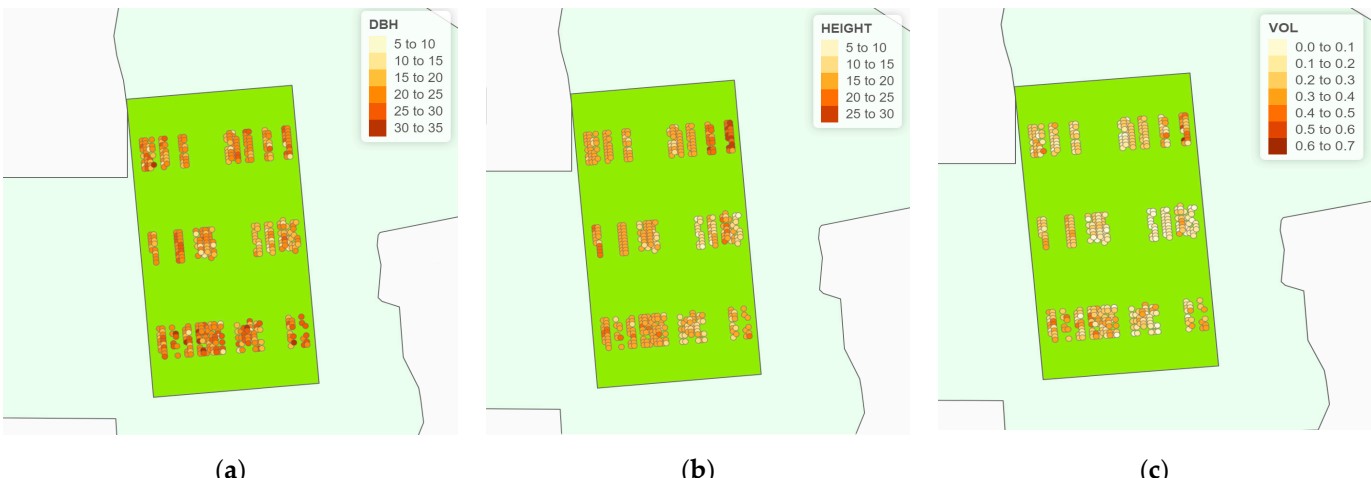

**Figure 8.** Sample stand spread of (**a**) DBH, (**b**) height, and (**c**) volume in the teak plots of 7-year-old Solomon Island-derived clones in block 96G at Brumas Camp, Tawau, Sabah.

The topology map shown in Figure 1 shows the elevation, which may affect tree growth. Thus, the changes in the density of the tree growth throughout the block depend on spatial variables such as elevation differences in the topography that may affect the amount of sunlight received, as there may be shadows from a hill, or even the chemical distribution of the soil properties. This indicates that spatial dependence affects teak tree growth.

## 4. Discussion

The analysis using Moran's I supports the hypothesized model that there is a strong spatial dependence between the physical parameters, especially height and volume. The plots of the Moran scatterplots in Figures 3 and 4 further solidify these findings by showing a positive trend for all the physical parameters for both years. These plots show a steady increase as the tree aged over the course of a year. Sabah is close to the equator, where there should be approximately 12 h of sunlight a day all year round, and this amount of sunlight affects the height of these trees. The topology map in Figure 1 shows that the block is located on sloped land, which may have affected the amount of direct sunlight received in some areas and could account for the unmeasured variable that causes height differences. However, when it comes to DBH, many environmental influences could be the cause of high variance. The soil composition, climate variation, or immediate competitive environment may have contributed as a variable affecting the DBH of the tree.

These two attributes ultimately make the formula to determine the volume of the tree, which is an argument on its own. A tree may be tall and have a small DBH, or it may be short and have a large DBH, and both situations would affect the outcome of the volume calculation. The formula does not include the mortality risk or diseases that the trees might develop over the years that could cause hollowed logs or rotten insides, but this information is only known once the tree is felled. There have been studies that have suggested increasing the accuracy of volumetric models. However, the calculated Moran's I values and the Moran scatterplots prove that spatial variability must also be considered.

The semivariograms showed how the spatial dependency changed from the 6th year to the 7th year in every physical parameter examined. The plots showed an increase in the semivariance, indicating an increase in spatial dependency compared to the parameters in the 6th year to the 7th year, satisfying the second hypothesis of this study. Thus, spatial dependency should increase as the teak trees age. As reported by [5] and studies such as [37–39], as the tree matures, spatial dependencies decline towards zero. However, these studies were for *Eucalyptus pilularis* [5], silver maple tree [37], and forest trees of similar ages [38] and forest structures [39]. There is little literature to support teak's spatial dependence as the tree matures, meaning that there is a need for further research.

Thematic maps are visual representations of the collected samples. Large sample sizes help to visualize the possible outcomes of how the tree's physical parameters are distributed throughout the block. The plots in this study further solidify both hypotheses, as a non-random spatial distribution was observed for both years and for all parameters.

## 5. Conclusions

This work serves as a preliminary study to further investigations and mathematical growth models considering spatial dependence, and ascertains the need to identify the existence of spatial dependence, as the succeeding growth models relies on information about the spatial dependency. The study proves that spatial dependence exists and affects the growth of the teak trees specific to Solomon Island-derived clones in Tawau, Sabah. Furthermore, the magnitude of dependence towards the spatial attributes showed an increase within a year of growth. This provides us with the understanding that competition either intensifies or abates as teak trees age. Since felled trees are not necessary to estimate growth, there is a clear need to further the research to see whether it is in agreement with previous findings stating that the magnitude and even signs of spatial dependencies change through stand development based on the relative strength of two spatial processes: competition and micro-site development [30,40,41].

The teak stands specific to Solomon Island-derived clones in Tawau, Sabah, Malaysia, have long been under study by pioneers in the field [21,22] striving to develop clones to increase the productivity of site management. This is because foresters are the main influencers of the development of their forest stands through a number of decisions and silvicultural treatments [42]. Hence, mathematical models act as verbal models to understand a tree's growth. These models complement our knowledge via their spatial data and further our understanding of tree growth. As discussed in [5], previous studies such as [4,43] found that a tree's growth is competitive but that it escapes the growth model and appears in the residuals, where its presence threatens estimation efficiency, ultimately affecting the validity of a model's inferences. The need to examine the model's residuals to identify whether the spatial dependencies result in negative or positive spatial dependence is crucial, as well. This provides a sound reason to continuously improve all aspects of growth model methodologies by incorporating spatial dependence [5].

The findings of this study will not only aid in the management of teak trees in Tawau, Sabah, Malaysia, but will also inspire and innovate studies on other types of forest trees. The importance of considering spatial dependence in every mathematical model as an additional variable for better estimation efficiency and validity is indubitable.

**Author Contributions:** Conceptualization, J.J.K. and R.M.Y.; methodology, J.J.K. and R.M.Y.; software, J.J.K.; validation, R.M.Y.; formal analysis, J.J.K.; resources, Y.J. and M.L.; data curation, M.L.; writing—original draft preparation, J.J.K.; writing—review and editing, R.M.Y.; visualization, R.M.Y.; supervision. R.M.Y.; project administration, R.M.Y. All authors have read and agreed to the published version of the manuscript.

**Funding:** This research received no external funding.

**Data Availability Statement:** Not applicable.

**Acknowledgments:** We would like to acknowledge the research and development team of Sabah Softwood Berhad, Tawau, Sabah, Malaysia, for providing the data for this research.

**Conflicts of Interest:** The authors declare no conflict of interest in this study.

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
