# Peer review of "Specifying Spatial Dependence for Teak Stands Specific to Solomon Island-Derived Clones in Tawau, Sabah, Malaysia: A Preliminary Study"

_sustainability, doi:10.3390/su14106005_

Round 1

Reviewer 1 Report

Comments

  • Line 60: Ghosh et al. (2019) Environmental Monitoring and Assessment, 191(3), 786 can be added here.
  • Provide a reference for equation 1.
  • Line 117-118: Provide reference.
  • Line 139: The country name should also be mentioned.
  • Line 159-160: Did you record the coordinates of the trees as well.
  • Line 163: “…… data will be analysed”. I hope the data has been analysed already.
  • Line 167: What was the accuracy of the GPS?
  • Line 176: “……….. rough flow chart”. Why are you referring it as ‘rough’?
  • The conclusion section is missing. Should be added.
  • The utility of this kind of study from a sustainability point of view should be discussed.

Author Response

Dear Referee,

Please see the attached file to see our responses.

Thank You

Rossita

Reviewer 2 Report

The present article is approaching the issue of teak trees from Solomon Islands.

First of all, the title must be simplified/ clarified. In the present form is quite unclear.

In the abstract, the issue of spatial dependence (in relation with what?) needs also a clearer approach. 

Please pay attention on how the citations are made in the text. Please check the MDPI rules for this.

Figures 3, 4, 5 and 6 needs to be improved in terms of quality since, in this present form, they are hardly readable.

The discussions section is under-developed. Which is the main conclusion of your work?

The article needs to follow a major proof-reading process. Due this problem, the methodology used in this work is not clearly presented.

It is not clear if the authors followed all the 6 steps mentioned in figure 2 to achieve the article objectives. Please clarify this aspect.

The article seems also to be focused on a local situation. It is not clear if the results/ conclusions of this work can be applied/ are valid for other areas also.

The reference list is not written according to MDPI rules.

To conclude:

  • the main objective of this article, the used methodology, the results and the main conclusion should be better correlated.
  • the methodology is not clear, mainly the considered algorithm
  • Please clearly mention the hypothesis considered in your study
  • Please perform a proof-reading of you article before re-submittion

Author Response

(The authors gave the same response as above.)

Round 2

Reviewer 2 Report

The authors succeed to provide clear and relevant answers to the questions mentioned after the first review.

The revised manuscript can be accepted as it is.